# Changes in health-related quality of life in young-old and old-old patients undergoing elective orthopedic surgery: A systematic review

Yun Jin Chen[1‡], Justine Lau[2‡], Yasmin Alhamdah[3,4], Ellene Yan[3,4], Aparna Saripella[4], Marina Englesakis[5], David He[6], Frances Chung[3,4]*

1 Queen's University School of Medicine, Kingston, ON, Canada, 2 Michael G. DeGroote School of Medicine, McMaster University, Hamilton, ON, Canada, 3 Institute of Medical Science, Temerty Faculty of Medicine, University of Toronto, Toronto, ON, Canada, 4 Department of Anesthesia and Pain Management, Toronto Western Hospital, University Health Network, University of Toronto, Toronto, ON, Canada, 5 Library & Information Services, University Health Network, Toronto, ON, Canada, 6 Department of Anesthesiology and Pain Medicine, Mount Sinai Hospital, Sinai Health, Toronto, Ontario, Canada

‡ YJC and JL share first co-authorship on this work.
* frances.chung@uhn.ca

## Abstract

### Background

There is a significant gap in research exploring changes in postoperative health-related quality of life (HRQoL) among patients aged 65 years and older undergoing hip or knee arthroplasty.

### Objectives

To investigate the variations in HRQoL improvement, as evaluated by patient-reported outcome measures following total hip arthroplasty, total knee arthroplasty, and partial knee arthroplasty between the young-old and old-old adults.

### Methods/Design

We searched six online databases (including MEDLINE, Embase) from their inception dates to May 15, 2023. We included studies using a validated HRQoL assessment tool to evaluate changes in HRQoL in patients aged ≥65 years undergoing hip or knee arthroplasty. These include the EuroQol five-dimension (EQ-5D), Short Form 36 (SF-36) and Short Form 12 (SF-12). The primary outcomes were postoperative HRQoL changes between young-old (65–74 years) and old-old groups (≥75 years). The secondary outcomes included complications, length of stay, and mortality.

### Results

The search yielded 12,229 articles; twelve studies (n = 103,613) were included. Studies using EQ-5D found no significant differences between young-old and old-old patients after

**Data Availability Statement:** All relevant data are within the manuscript and its Supporting Information files.

**Funding:** The author(s) received no specific funding for this work.

**Competing interests:** I have read the journal's policy and the authors of this manuscript have the following competing interests: FC reports research support from the Ontario Ministry of Health Innovation Grant, ResMed Foundation, University Health Network Foundation, Consultant to Takeda, and STOP-Bang Questionnaire proprietary to University Health Network. This does not alter our adherence to PLOS ONE policies on sharing data and materials.

hip and knee arthroplasty. Analyses of SF-36 and SF-12 scales showed no significant age-related differences in postoperative improvements in physical and mental health. Our review of four studies that included multivariable analyses revealed inconsistent associations between age and EQ-5D. Comparisons between the young-old and old-old age groups in postoperative complications, hospital length of stay, and mortality revealed no associated age-related changes in HRQoL.

## Conclusions

The young-old and old-old patients exhibited comparable improvement in HRQoL following hip or knee arthroplasty. The older patients did not have higher postoperative complications rates, longer hospital length of stay, and increased mortality. While chronological age should be considered when planning hip and knee arthroplasty, greater emphasis should be placed on assessing the comorbidities and functional status of patients.

## Introduction

In 2019, 743,327 total hip and knee arthroplasties were performed in the United States alone. By 2040, the annual number of these procedures is projected to increase to approximately 1.9 million, given that individuals aged 65 years and older will constitute a quarter of the population and the prevalence of osteoarthritis is increasing [1–3]. The primary aims of these surgeries are to reduce pain, and improve joint function and quality of life. Domains such as health-related quality of life (HRQoL) are frequently assessed using patient-reported outcomes measures to determine the success rate of arthroplasty and post-surgical satisfaction [4]. Although total hip arthroplasty (THA) and total knee arthroplasty (TKA) are very effective surgeries, the rate of dissatisfaction ranges from 7 to 20% [5–7]. Determining which patients will benefit most from hip and knee arthroplasty is important when considering the rising patient load, surgical needs, and resource allocation [2].

One risk factor that has been suggested to impact HRQoL following hip or knee arthroplasty is age [8, 9]. Data from the World Population Prospects predicts that the proportion of older adults (≥ 65 years of age) will reach 1.5 billion by 2050 [10]. Thus, it is important to understand how age impacts HRQoL after hip and knee arthroplasty.

Various approaches on how to define "young-old" and "old-old" have been explored, with no current consensus on age ranges [11–17]. Most studies have designated individuals aged 65–74 years as young-old and those aged 75 years and above as old-old [11–13, 16, 17]. To date, there are limited studies examining whether HRQoL differs between young-old and old-old adults following hip or knee arthroplasty and partial knee arthroplasty (PKA). The current evidence in the literature is contradictory, with some studies reporting similar benefits regardless of age and others finding differences in HRQoL improvement with varying age [18–23]. With the diversity of age in the older population, there is a great need to determine clinically relevant differences in HRQoL outcomes between young-old and old-old adults undergoing hip and knee arthroplasty [24].

This systematic review aims to determine whether there are differences in HRQoL following THA, TKA, and PKA between the young-old and old-old adults. The secondary objectives are to assess how age impacts other outcomes such as postoperative complications, length of stay (LOS), and mortality. These findings will contribute to the informed selection of patients

undergoing hip and knee arthroplasty, thereby promoting improved HRQoL, and optimizing resource allocation in an aging population.

## Methods

### Study registration

The protocol for this systematic review is registered in the International Prospective Register of Systematic Reviews (PROSPERO) (CRD42023439985) and follows the Preferred Reporting Items for Systematic Review and Meta-Analyses (PRISMA) reporting guidelines [25].

### Search strategy

The search strategy was developed and implemented by an experienced information specialist (ME). MEDLINE, MEDLINE ePubs and In-Process Citations (daily), Embase, Cochrane Central Register of Controlled Trials (CCTR), Cochrane Database of Systematic Reviews (CDSR), and Scopus were searched from their inception dates to May 15, 2023.

Preliminary searches were completed, and full text literature was reviewed for potential keywords and appropriate controlled vocabulary terms (such as Medical Subject Headings for Medline and EMTREE descriptors for Embase). The Yale MeSH Analyzer was used to assess target citations. The search strategy concept blocks were built on the topics of: "total knee arthroplasty" OR "total hip arthroplasty" AND "older adult" OR "elderly" AND "EQ-5D" OR "SF-36" OR "SF-12", with each component being expanded with controlled vocabularies, text word terms, and synonyms. Results were limited to English language, humans, and adults. Continued literature surveillance was conducted throughout the year until November 2023. The MEDLINE search strategy is provided in supplementary material.

### Study selection

Two reviewers (YJC, JL) independently screened the titles and abstracts of identified articles using Covidence [26]. Full-text review was done independently by two reviewers (YJC, JL). Conflicts regarding inclusion of abstracts and full-text articles were resolved through discussion with a third reviewer (YA). Any further disagreements were resolved through discussion with the senior author (FC). The inclusion criteria were: 1) patients aged ≥65 years, 2) undergoing THA, TKA, or PKA, 3) evaluated preoperatively and postoperatively with a validated HRQoL assessment tool 4) randomized controlled trials (RCTs), prospective observational studies, case control studies, cross-sectional studies, 5) sample size ≥100, and 6) English language. Case reports, retrospective studies, and systemic reviews were excluded. Studies with bilateral joint arthroplasty were also excluded.

### Data extraction

Data from the included full-text studies were independently extracted by two reviewers (YJC, JL) using an Excel spreadsheet. Extracted data consisted of author, publication year, country, study design, participants' demographics, surgery type, sample size, mean age, sex, mean pre- and postoperative HRQoL scores and reported change, timeline of HRQoL measurement, and incidence of secondary outcomes (post-operative complications, hospital LOS, and mortality). Studies providing only median (IQR) values were converted to mean (SD) to facilitate quantitative analysis.

## Quality of study assessment

Study quality was evaluated by two independent reviewers (YJC, EY) based on the study design. Disagreements between the two independent assessors were resolved through discussion, with consensus from another reviewer (YA). For prospective cohort and case control studies, quality was evaluated in accordance with the Meta-Analyses of Observational Studies in Epidemiology (MOOSE) checklist [27] and the Newcastle-Ottawa scale (NOS) [28]. The MOOSE checklist evaluated the following domains: clearly identified population, clear outcomes and outcome assessment method, selective loss of patients in the follow-up, and identification of confounding variables. The NOS rated the selection, comparability, and exposure of cohorts, with a maximum score of nine indicating higher quality. Study quality was categorized as good if the assigned score was ≥8, fair for scores between five and seven, or poor for scores below five.

## Measures of HRQoL

To assess HRQoL and mental health, EuroQol five-dimension (EQ-5D), Short Form 36 (SF-36), and Short Form 12 (SF-12) are the validated and most frequently used tools [29, 30] (Table 1). EQ-5D is a generic and preference-weighted measure that consists of a two-part patient-administered questionnaire [31, 32]. It evaluates five dimensions: mobility, self-care, daily activities, pain/discomfort, and anxiety/depression. Responses are converted into a single measure of health utility (EQ-5D index score) using certain weights that are country-specific; a score of less than 0 indicates a state worse than death, 0 indicates death, and 1 represents perfect health [33].

The Short Form 36 (SF-36) is a widely used, validated, multipurpose, self-reported health survey that consists of eight dimensions (physical function, role–physical, bodily pain, general health, vitality, social function, role–emotional, and mental health) and two summary scores (physical and mental components) [34]. Each dimension score ranges from 0 to 100, with higher scores indicating a better quality of life. The mean Physical Component Summary (PCS) and Mental Component Summary (MCS) for the general American population are 50 with a standard deviation (SD) of 10 [35].

The Short Form 12 (SF-12) is developed from the SF-36 and covers the same eight domains as the SF-36 [36]. Similarly, it generates two summary scores: SF-12 physical and SF-12 mental scores. The scales are also transformed to achieve a mean of 50 (± 10) in the general American

**Table 1. HRQoL measures and their corresponding range of scores and indications.**

| Tool | Range of Scores | HRQoL Indication |
|---|---|---|
| **EQ-5D**[a] | <0–1 | 1 = full health<br>0 = state as bad as being death<br><0 = state regarded as worse than being dead |
| **SF-36** | 0–100 | 0 = worst possible health<br>100 = best possible health<br>MCS ≤42 = depression<br>PCS ≤50 = below average health |
| **SF-12** | 0–100 | 0 = worst possible health<br>50 = average score for population<br>100 = best possible health<br>MCS ≤42 = depression<br>PCS ≤50 = below average health |

*Abbreviations*: EQ-5D: EuroQol five-dimension; SF-36: Short Form 36; SF-12: Short Form 12; PCS: physical component score; MCS: mental component score

[a] Limit of score range depends on algorithm used to calculate EQ-5D index score.

population [37]. Given that PCS and MCS use norm-based scoring, scores higher (or lower) than 50 indicate better (or worse) physical (or mental) health relative to the population of the given sample [38]. It is suggested that a score of 42 or less on the MCS may indicate 'clinical depression' [37, 39]. Overall, it is a more practical research tool as it has substantially fewer questions in the survey.

## Data analysis

Qualitative analyses were carried out for the included studies. Study characteristics, patient demographics, pre-and postoperative HRQoL scores at common time points (3-, 12- and 24-months), and secondary outcomes were summarized descriptively. While consensus is lacking in the optimal sub-classification of older adults into different age groups [11–17], several studies designated individuals aged 65–74 years as young-old and those aged 75 and above as old-old [11–13, 16, 17]. Accordingly, we defined "young-old" as individuals aged 65–74 years and "old-old" as those aged ≥75 years. In a few instances where these age groups were not used [18–21, 40, 41], we classified the younger group (<80 years) as "young-old" and the older age group (≥80 years) as "old-old".

## Results

### Study selection and characteristics

A total of 12,229 articles were identified from the databases (Fig 1). After duplicate removal, title, and abstract screening, 207 full-text articles were evaluated for eligibility. Of these articles, 12 studies met the inclusion criteria and were included for qualitative analysis [18–23, 40–45]. The demographic data and study characteristics of the 12 included studies are summarized in Table 2 [18–23, 40–45]. A total of 103,613 patients were included in the 12 studies, with a mean age ± SD of 67.5 ± 10.2 years old and 54.1% female. Studies from seven different countries were included: United States (n = 3), United Kingdom (n = 3), Canada (n = 2), Denmark (n = 1), Germany (n = 1), Taiwan (n = 1), and Spain (n = 1). Ten studies were cohort studies and two were prospective case control studies [42, 43]. Five studies examined THA [18, 20, 21, 44, 45], two TKA [19, 43], two PKA [40, 42], two THA and TKA [22, 41], and one TKA and PKA [23].

### Quality assessment

Quality assessment is presented in S1 and S2 Tables of S1 File. In adherence to the MOOSE checklist, all studies clearly identified their study populations, outcomes, and outcome assessment methods. However, one study did not report the loss of patients during follow-up [22]. All included articles identified important confounding variables, such as demographic information. The NOS scores ranged from five to eight for prospective cohorts and four to five for case-control studies. Notably, studies scored well in domains of representativeness of the exposed cohort, selection of the non-exposed cohort, absence of the outcome of interest at study onset, cohort comparability, and duration of follow-up for outcomes to occur. Studies scored somewhat well in the ascertainment of exposure and adequacy of follow-up but poorly in outcome assessments. Overall, there was heterogeneity in the quality of the prospective studies, ranging from fair to good.

### Comparing HRQoL change between young-old and old-old groups

**Measures of health-related quality of life.** We grouped the studies by the measures of HRQoL: EQ-5D, SF36, and SF-12 according to young-old and old-old groups (Tables 3 and 4). The pre- and postoperative HRQoL scores at various time points were reported. The

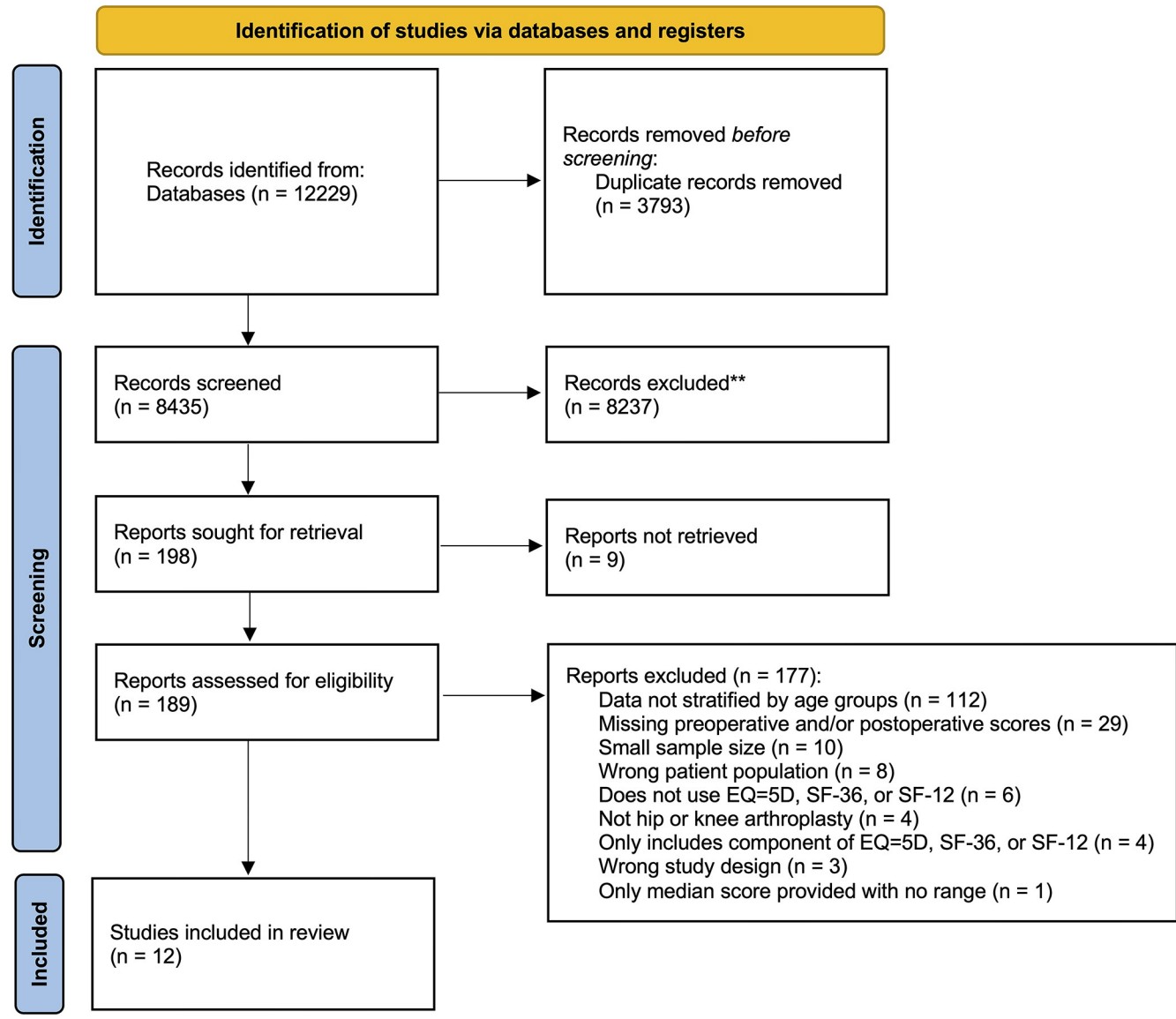

**Fig 1. PRISMA flow diagram.** PRISMA, Preferred Reporting Items for Systematic Review and Meta-Analyses.

change in scores between the different older adult age groups were provided in six studies (S3 and S4 Tables in S1 File) [19–21, 23, 42, 44]. Most of the studies have reported the significance in terms of minimal clinically important difference, which indicates the smallest change in HRQoL score that a patient considers as meaningful improvement.

**Change in HRQoL measured by EQ-5D.** Seven studies (n = 93,183) reported preoperative and postoperative EQ-5D values for young-old and old-old surgical patients with various time points (Table 3) [18, 20–23, 40, 45]. The commonly evaluated postoperative time points were at 3-, 12-, and 24-months. At the 3-month postoperative time point, a prospective cohort study by Aalund et al. reported that in a 1,610 patient sample, the old-old group (≥80 years old) demonstrated greater improvements in EQ-5D scores than the young-old group (70–79 years old) after total hip arthroplasty (P<0.001) [18]. To further support these findings, this same study also performed a multivariate analysis which reported a non-linear, positive

**Table 2. Demographics and characteristics of surgical patients.**

| First author, year, country | Study design | Type of surgery | Total patients (n) | Age (years): mean ± SD | Females: % |
|---|---|---|---|---|---|
| *EQ-5D* | | | | | |
| Aalund, 2017, Denmark, [18] | PC | THA | 1,610 | 68.9 ± 10.2 | 47.8 |
| Anderson, 2022, Germany, [20] | PC | THA | 300 | 74.0 ± 7.0 | 56.7 |
| Clement, 2022, UK, [45] | PC | THA | 200 | 69.9 ± 9.5 | 57.5 |
| Gordon, 2014, Sweden, [21] | PC | THA | 34,519 | 69.0 ± 10.0 | 57.0 |
| Miao, 2018, Taiwan, [22] | PC | THA, TKA | 105 | 71.8 ± 9.9 | 68.6 |
| Tadros, 2018, UK, [40] | PC | PKA | 395 | 70.7 ± 7.1 | 45.6 |
| Williams, 2013, UK, [23] | PC | PKA, TKA | 2,456 | 71.4 ± 9.1 | 60.8 |
| *SF-36* | | | | | |
| Alentorn-Geli[a], 2013, Spain, [19] | PC | TKA | 391 | 72.5 ± 6.7 | 77.5 |
| Ayers, 2022, USA, [44] | PC | THA | 7,934 | 64.5 ± 10.7 | 56.5 |
| Goh, 2020, USA, [43] | PCC | TKA | 1,188 | 75.6 ± 6.3 | 79.9 |
| Jones[a], 2001, Canada, [41] | PC | THA, TKA | 454 | 71.0 ± 7.9 | 60.1 |
| *SF-12* | | | | | |
| Goh, 2021, USA, [42] | PCC | PKA | 176 | 73.3 ± 6.3 | 48.3 |

Abbreviations: PC: Prospective cohort; PCC: Prospective case control; TKA: Total Knee Arthroplasty; THA: Total Hip Arthroplasty; PKA: Partial Knee arthroplasty.

[a] likely includes patients <65 years

association between age and postoperative EQ-5D improvement at 3- and 12-months (Table 5). In other words, older patient age was associated with greater changes in EQ-5D scores. More specifically, an age gap of at least 12 and 15 years between young-old (70–79 years) and old-old (≥80 years) patients was necessary to show clinically relevant EQ-5D differences between the two groups at the 3- and 12-month follow-up, respectively [18].

**Table 3. Pre- and postoperative EQ-5D values in hip and knee arthroplasty patients.**

| First author, year | Age group (years) | Patients in each age group (n) | Preoperative EQ-5D score: mean ± SD | Postoperative EQ-5D score: mean ± SD | | | Differences in HRQoL change between age groups |
|---|---|---|---|---|---|---|---|
| | | | | 3-month | 12-month | 24-month | |
| Aalund, 2017, [18] | 70–79 | 548 | 0.63 ± 0.20 | 0.89 ± 0.14 | 0.90 ± 0.14 | NR | 3-m (P<0.001) |
| | ≥80 | 169 | 0.52 ± 0.23 | 0.81 ± 0.15 | 0.82 ± 0.15 | | 12-m (P<0.001) |
| Anderson, 2022, [20] | 70–79 | 100 | 0.46 ± 0.23 | NR | 0.92 ± 0.20 | NR | n.s. |
| | ≥80 | 100 | 0.47 ± 0.24 | | 0.79 ± 0.29 | | |
| Clement, 2022, [45] | 65–74 | 67 | 0.32 ± 0.11 | 0.82 ± 0.17 | 0.84 ± 0.19 | 0.660 ± 0.140 | NR |
| | ≥75 | 64 | 0.46 ± 0.23 | 0.63 ± 0.13 | 0.79 ± 0.22 | 0.600 ± 0.100 | |
| Gordon, 2014, [21] | 71–80 | 11,788 | 0.76 ± 0.13 | NR | 0.90 ± 0.12 | NR | NR |
| | ≥81 | 4,784 | 0.73 ± 0.13[a] | | 0.87 ± 0.14[a] | | |
| Miao, 2018, [22] | 66–75 | NR | 0.19 ± 0.37 | 0.71 ± 0.27 | NR | NR | NR |
| | >75 | | 0.23 ± 0.35 | 0.59 ± 0.36 | | | |
| Tadros, 2018, [40] | 70–79 | 149 | 0.50 | NR | 0.80 | 0.80 | NR |
| | 80–89 | 58 | 0.50 | | 0.80 | 0.80 | |
| Williams, 2013, [23] | 65–74 | 937 | 0.43 ± 0.32 | NR | NR | 0.76 ± 0.27 | n.s. |
| | ≥75 | 922 | 0.45 ± 0.31 | | | 0.73 ± 0.25 | |

Abbreviations: THA: Total Hip Arthroplasty; TKA: Total Knee Arthroplasty; NR: Not Reported; n.s.: Not Significant; HRQoL: Health-Related Quality of Life; EQ-5D: EuroQol- 5 Dimension.

[a] calculated weighted mean

**Table 4. Pre- and postoperative SF-36 or SF-12 values in hip and knee arthroplasty patients.**

| First author, year | Age group (years) | Preoperative MCS score: mean ± SD | Postoperative MCS score: mean ± SD | | | Preoperative PCS score: mean ± SD | Postoperative PCS score: mean ± SD | | | Differences in HRQoL change between age groups |
|---|---|---|---|---|---|---|---|---|---|---|
| | | | 6-month | 12-month | 24-month | | 6-month | 12-month | 24-month | |
| *SF-36* | | | | | | | | | | |
| Alentorn-Geli[a], 2013, [19] | <80<br>≥80 | 46.6 ± 15.0<br>49.9 ± 14.0 | NR | 47.9 ± 13.0<br>48.6 ± 13.0 | NR | 32.1 ± 7.0<br>30.8 ± 9.0 | NR | 44.2 ± 9.0<br>39.9 ± 7.0 | NR | PCS: n.s.<br>MCS: n.s. |
| Ayers, 2022, [44] | 65–74<br>≥75 | 52.5 ± 11.8<br>51.3 ± 11.8 | NR | 55.4 ± 8.7<br>54.1 ± 9.8 | NR | 31.7 ± 8.6<br>31.2 ± 8.3 | NR | 45.6 ± 9.9<br>42.3 ± 10.3 | NR | NR |
| Goh, 2020, [43] | 65–74<br>≥80 | 52.4 ± 10.3<br>52.2 ± 10.3 | 54.1 ± 10.4<br>55.3 ± 9.9 | NR | 55.5 ± 10.2<br>55.0 ± 10.2 | 29.5 ± 10.5<br>29.5 ± 9.9 | 45.2 ± 11.4<br>42.3 ± 11.3 | NR | 48.1 ± 10.0<br>45.2 ± 11.1 | NR |
| Jones[a], 2001, [41] | <80<br>≥80 | THA:<br>54.0 ± 10.0<br>50.0 ± 14.0<br>TKA:<br>50.0 ± 11.0<br>51.0 ± 11.0 | 54.0 ± 10.0<br>51.0 ± 14.0<br>53.0 ± 11.0<br>51.0 ± 11.0 | NR | NR | 26.0 ± 6.0<br>22.0 ± 5.0<br>26.0 ± 8.0<br>25.0 ± 7.0 | 38.0 ± 11.0<br>35.0 ± 11.0<br>35.0 ± 10.0<br>32.0 ± 10.0 | NR | NR | NR |
| *SF-12* | | | | | | | | | | |
| Goh, 2021, [42] | 65–74<br>≥80 | 55.0 ± 8.2<br>52.8 ± 9.7 | NR | NR | 53.8 ± 8.0<br>53.1 ±14.2 | 35.4 ± 8.7<br>34.4 ± 9.6 | NR | NR | 44.9 ± 9.2<br>39.4 ± 14.1 | n.s. in both PCS (P = 0.263) and MCS (P = 0.981) |

Abbreviations: THA: Total Hip Arthroplasty; TKA: Total Knee Arthroplasty; PCS: physical component score; MCS: mental component score; NR: Not Reported; n.s.: Not Significant; HRQoL: Health-Related Quality of Life

[a] likely includes patients <65 years

However, two other studies had contradicting results [20, 23]. A prospective cohort study on hip arthroplasty by Anderson et al. found no significant differences in postoperative EQ-5D score changes between young-old (70–79 years) and old-old (≥80 years) patients at the 12-months follow-up in a sample of 300 patients [20]. Forty-four percent of the old-old (≥80 years) and sixty percent of the young-old (70–79) patients experienced postoperative changes in EQ-5D scores that achieved minimal clinically important difference thresholds; however, it was not reported if these percentages are statistically different [20, 46]. Moreover, another prospective cohort study by Williams et al. on knee arthroplasty found no significant differences

**Table 5. Multivariate analysis of factors associated with postoperative EQ-5D change.**

| First author, year | Postoperative timepoint (months) | Association between age and EQ-5D change | Other coefficients for EQ-5D change |
|---|---|---|---|
| Aalund, 2017, [18] | 3<br>12 | 0.0026 (P<0.001)<br>0.0020 (P = 0.001) | Preoperative EQ-5D score:<br>-0.841 (P<0.001)<br>-0.804 (P<0.001) |
| Gordon, 2014, [21] | 12 | Negative, non-linear association | Preoperative EQ-5D score: -3.9 |
| Miao, 2018 [22] | 6 | n.s. | Women: 7.613 (P<0.05)<br>No comorbidities: 3.259 (P<0.05)<br>Joint operated on: 15.490 (P<0.05)<br>Employment status, living status, physical barriers at home: n.s. |
| Williams, 2013, [23] | 6<br>24 | Negative, linear association:<br>P = 0.013<br>P = 0.033 | NR |

Abbreviations: EQ-5D: EuroQol- 5 Dimension; NR: Not Reported; n.s.: Not Significant.

in EQ-5D improvements of young-old (65–74 years old) and old-old (≥75 years old) groups in a sample of 2,465 patients [23]. The multivariate analysis showed a negative relationship with a linear trend between age and EQ-5D change at both 6- and 24-months follow up. As a result, older age was associated with decreased postoperative gains in EQ-5D score [23]. Likewise, Gordon et al. revealed a negative association between age and EQ-5D improvement following hip arthroplasty [21]. However, at the 12-month follow-up, a non-linear trend was noted, with EQ-5D changes generally unaffected by age until the late sixties, beyond which increasing age had a negative effect [21].

Four studies did not report statistical analysis on HRQoL changes in young-old and old-old patients following hip or knee arthroplasty [21, 22, 40, 45]. However, Clement et al. did report that both young-old (65–74 years) and old-old (≥75 years) groups had significant improvement in EQ-5D scores following hip arthroplasty (P<0.001). In addition, there were no significant differences in postoperative EQ-5D scores between young-old and old-old patients at 3-, 12-, and 24-months follow up [45]. These findings suggest that all ages of older adults stand to benefit from hip arthroplasty. Furthermore, Gordon et al. reported that most patients improved their EQ-5D score regardless of age. However, the researchers also found that 13% of old-old patients (≥80 years) failed to improve their EQ-5D score [21]. It is important to acknowledge that this percentage represents a minority of their patient population; the decline in improvement may be attributed to the natural age-related deterioration of HRQoL that is not hip-related, thus reducing the beneficial impact of hip arthroplasty. Moreover, Miao et al. performed a multivariate analysis of their patient sample which showed age was not significantly associated EQ-5D scores [22]. Lastly, Tadros et al. found no significant differences in postoperative EQ-5D scores between young-old (70–79 years) and old-old (80–89 years) patients following partial knee arthroplasty. Moreover, both young-old and old-old patients in this study had significant postoperative improvements in their postoperative EQ-5D score at the 12-month follow up. These improvements were also found to be sustained until 24-months follow up for both age groups [40].

**Change in HRQoL measured by SF-36 and SF-12.** Five studies (n = 10,254) reported SF-36 values [19, 41, 43, 44], and one study (n = 176) reported SF-12 values for young-old and old-old groups [42] (Table 3B). The most commonly assessed postoperative time points were at 12- and 24-months. Alentorn-Geli et al. investigated changes in SF-36 score after total knee arthroplasty for 439 patients in Spain. In this study, they observed no significant differences in the PCS and MCS between patients aged ≥80 years vs <80 years at 12-months [19]. This suggests that the improved quality of life between the patient age groups were comparable. Similarly, Goh et al. investigated SF-12 score changes after partial knee arthroplasty in 44 octogenarians who were matched with 132 patients aged 65–74 in USA. They reported no significant differences in PCS and MCS between young-old (65–74 years) and old-old (≥80 years) patients at 24-months [42]. More specifically, a lower percentage of octogenarians attained the minimal clinically important difference for SF-12 physical scores, but this was not significantly different compared with the younger age group [42]. This suggests that the young-old (65–74 years) and old-old (≥80 years) patients had a similar improvement in knee function and well-being postoperatively. Another single-center study prospectively collected data of 594 octogenarians who were matched with 594 patients aged 65–74 who underwent total knee arthroplasty [43]. In contrast, they found that at both 6-months and 2 years postoperatively, a significantly lower proportion of the old-old (≥80 years) patients achieved the minimal clinically important difference for SF-36 PCS (p<0.001, p = 0.010). Despite the old-old (≥80 years) patients showing relatively lesser improvements in quality of life compared to the young-old controls (65–74) after knee arthroplasty, all old-old patients still achieve significant functional gains compared to their preoperative status.

Jones et al. studied the change in SF-36 scores in 197 patients following hip arthroplasty and 257 patients who underwent knee arthroplasty in Canada. They found that both the young-old (<80 years) and old-old (≥80 years) age groups showed large effect sizes in physical function 6-months postoperatively [41]. The effect sizes in mental health dimensions after 6 months were comparatively modest for both age groups, yet the values fell within the range observed in the general population [41]. Ayers et al. investigated a multicenter cohort of 7934 patients that underwent hip arthroplasty and noted changes in SF-36 mental and physical well-being after 12 months. The difference in 12-months postoperative PCS was slighter higher in the young-old (65–74 years) patients, and although statistically significant (P<0.0001), did not reach the minimal clinically important difference [44]. This indicates that the difference was not clinically significant and thus, the perceived HRQoL improvements between the different age groups were similar.

**Multivariate analysis of factors associated with EQ-5D change.**   Besides age, four studies performed a multivariate analysis to assess how various factors such as sex, comorbidities, and preoperative EQ-5D score impact changes in EQ-5D following hip or knee arthroplasty (Table 5) [18, 21–23]. Two studies reported negative associations between preoperative EQ-5D and overall EQ-5D change [21, 23]. Aalund et al. found a higher preoperative EQ-5D score had a negative effect on EQ-5D improvement at 3- and 12-months follow-up [18]. Similarly, Gordon et al. reported greater preoperative EQ-5D scores was associated with decreased EQ-5D improvements at 12-months follow up. Furthermore, both studies showed that the impact of preoperative EQ-5D on HRQoL improvement was greater than age itself [18, 21].

In addition, various other factors affecting EQ-5D change were analyzed by Miao et al. Being female, having no comorbidities, and having a knee arthroplasty instead of hip arthroplasty were found to have positive associations with EQ-5D improvement (P<0.05) [22]. However, employment status, living status, physical barriers at home were found to have no significant impact on EQ-5D change.

## Postoperative outcomes between young-old and old-old groups

**Postoperative complications.**   Five studies (n = 1,525) reported postoperative complications following hip or knee arthroplasty (Table 6) [20, 40–42, 45]. Most of these studies showed no significant differences for orthopedic complications, need for revisions or reoperations, dislocations, wounds, deep vein thrombosis, and urinary tract infection between young-old and old-old age groups [20, 40–42, 45]. Only one study showed significantly increased non-orthopedic complications following total hip arthroplasty in the older age group (≥80 years) [20].

**Hospital length of stay (LOS).**   Four studies (n = 1,540) reported LOS following hip or knee arthroplasty [19, 20, 40, 41] (Table 6) with variable findings. One study found no significant differences between age groups in hospital LOS following total hip arthroplasty, but significantly decreased hospital LOS in the older age group (≥80 years) following total knee arthroplasty (6.0 ± 2.0 vs 7.0 ± 2.0, P = 0.04) [41]. Conversely, another study reported significantly increased hospital LOS in the older age group (80–89 years) compared with the younger patients (70–79 years) following partial knee arthroplasty (4.5 ± 2.2 vs 3.9 ± 1.7 days, P = 0.006) [40].

**Postoperative mortality.**   Five studies (n = 1,525) reported mortality in older adults following hip or knee arthroplasty [20, 40–42, 45] (Table 6). One study investigating 2-year mortality following partial knee arthroplasty reported no significant differences between the young-old (70–79 years) and old-old (80–89 years) age groups [40]. Similarly, another study investigating 5-year mortality following partial knee arthroplasty also reported no significant differences between the young-old (65–74 years) and old-old age groups (≥80 years) [42].

**Table 6. Comparison of postoperative complications, length of stay, and mortality between age groups.**

| First author, year | Surgery Type | Age group (years) | Timeline | Postoperative complications: n (%) | | | | Hospital LOS: mean days ± SD | Mortality: n (%) | Differences between age groups |
|---|---|---|---|---|---|---|---|---|---|---|
| *EQ-5D* | | | | | | | | | | |
| Anderson, 2022, [20] | THA | 70–79 ≥80 | 1 month | Orthopedic 8 (8.0%) 5 (5.0%) | | Non-orthopedic 8 (8.0%) 13 (13.0%) | | 8.0 ± 4.0 9.0 ± 3.0 | 1 (1%) 0 (0%) | PC: orthopedic (n.s.) non-orthopedic (P = 0.033) LOS: NR Mortality: NR |
| Clement, 2022, [45] | THA | 65–74 ≥75 | 1 years 5 years | Dislocation 2 (3.8%) 3 (5.2%) | | | | NR | 0 (0%) 1 (1.7%) 1 (1.9%) 12.1% (7) | PC: n.s. Mortality: NR |
| Tadros, 2018, [40] | PKA | 70–79 80–89 | 2 years | Revision: 5 (3.3%) 3 (5.1%) | | | | 3.9 ± 1.7 4.5 ± 2.2 | 4 (2.87%) 3 (5.2%) | PC: n.s. LOS: P = 0.006 Mortality: n.s. |
| *SF-36* | | | | | | | | | | |
| Alentorn-Geli[a], 2013, [19] | TKA | <80 ≥80 | NR | NR | | | | 8.3 ± 0.8 8.5 ± 0.9 | NR | LOS: n.s. |
| Jones[a], 2001, [41] | TKA | <80 ≥80 | 6 months | THA TKA 4 (2.0%) 0 (0.0%) 5 (2.0%) 0 (0.0%) | Wounds 8 (5.0%) 1 (3.0%) 10 (5.0%) 0 (0.0%) | DVT 7 (4.0%) 5 (15.0%) 11 (5.0%) 3 (1.0%) | UTI | 7.0 ± 5.0 8.0 ± 4.0 7.0 ± 2.0 6.0 ± 2.0 | 2 (1%) 1 (2.6%) | PC: n.s. LOS: THA (n.s.) TKA (P = 0.04) Mortality: NR |
| *SF-12* | | | | | | | | | | |
| Goh, 2021, [42] | PKA | 65–74 ≥80 | 5 years | Reoperation 3 (2.3%) 0 (0) | Aseptic Revision 2 (1.5%) 2 (4.5%) | Septic Revision 1 (0.8%) 1 (2.3%) | Wounds 2 (1.5%) 2 (4.5%) | NR | (3%) (6%) | PC: n.s. Mortality: n.s. |

Abbreviations: LOS: Length of stay; PC: Postoperative complications; Total Hip Arthroplasty; TKA: Total Knee Arthroplasty; PKA: Partial Knee Arthroplasty; DVT: Deep Vein Thrombosis; UTI: Urinary Tract Infection; NR: Not Reported; n.s.: Not Significant

[a] likely includes patients <65.

## Discussion

The objective of our systematic review was to evaluate whether changes in HRQoL differ between young-old and old-old patients that underwent hip or knee arthroplasty. Improvement in the quality of life and the reason for performing hip or knee arthroplasty were similar across age ranges [19, 20, 22, 23, 42]. We found that even the old-old patients had measurable improvements in their HRQoL scores. One study reported a drop in postoperative EQ-5D scores at 24 months compared to 3 months, likely reflecting decreased activity levels in older patients, as the EQ-5D measures mobility and daily activities [45]. The lower activity level in older patients post-arthroplasty is likely related to aging-related physical health and social/ employment changes, rather than limitations from the surgery itself. While the study did not detail changes in EQ-5D scores over time, the authors reported clinically significant improvement in HRQoL post-operatively for all age groups sustained at all time points.

Therefore, age was not identified as a barrier to effective hip and knee arthroplasty. To the best of our knowledge, this is the first systematic review that directly compares the HRQoL change in the young-old and old-old patients following hip and knee arthroplasty and clarifies the controversial findings in this domain. Moreover, our secondary outcomes included evaluating differences in postoperative complications, hospital LOS, and mortality between different

older age groups. We found that there was no definite relationship between age and negative postoperative outcomes [19, 20, 40, 42, 45].

Aging is a complex phenomenon. While chronological age increases uniformly for the general population, biological age does not [47]. Older adults of a certain chronological age can differ in their physical and cognitive function. This discrepancy is logical, considering that chronological age is physical or mathematical whereas biological age can be defined as physiological or functional age. Biological age is associated with frailty, and frailty is the best indicator of age-related health decline, including mortality [48]. In fact, frailty has been shown to be predictive of post-operative complications, prolonged LOS, and mortality [49, 50]. The clinical implications suggest that relying solely on chronological age to determine whether patients would benefit from surgical intervention is a biased approach with inherent flaws. Older age should not be a contraindication for hip and knee arthroplasty; instead, comorbidities, preoperative muscle strength, and functional status should be assessed to predict postoperative function [8, 51].

A common assumption is that older patients are less likely to tolerate and benefit from surgeries [52]. This bias is important to address as the older population is growing rapidly. The global number of individuals aged 80 years or over is estimated to reach 426 million in 2050 [10]. Currently, this population disproportionately contributes to patients undergoing surgery. Consequently, it is crucial to inform older patients about the benefits and risks they may encounter [53]. Patients who do not receive hip or knee arthroplasty may have further worsening of their functional status with time [54]. Maintaining the ability to perform activities of daily living is a crucial component of HRQoL in older adults. The emphasis on preserving functional status by surgery should apply to all older adults [55].

Apart from effects on physical function, the older population exhibited a net improvement in postoperative mental scores measured by SF-12 and SF-36 [41–44]. This suggests that hip and knee arthroplasty not only improved functionality but may have additional mental benefits. A recent meta-analysis reported a global prevalence of around 32% for geriatric depression [56]. Physical illnesses, specifically immobility, was found to have a strong association with depression in old age [56]. Osteoarthritis of lower joints has been found to increase the relative risk of depression and anxiety [57]. While there are various contributors to geriatric depression, actively managing chronic pain through these surgeries may help to prevent depression [58]. Thus, it is important to recognize the positive impact that hip and knee arthroplasty have on both physical and mental wellbeing.

Besides age, this systematic review identified various other factors associated with HRQoL improvements following hip and knee arthroplasty. Preoperative EQ-5D score was consistently found to have a negative correlation with EQ-5D change where lower preoperative scores were associated with greater postoperative improvements in EQ-5D. Those with a low preoperative HRQoL, regardless of age, were found to have a greater postoperative improvement in HRQoL [18, 21]. EQ-5D has been noted to face challenges related to ceiling effects [59]. Consequently, there might be a limit to the extent the EQ-5D score can change following hip and knee arthroplasty. This may partly explain why individuals with a lower preoperative EQ-5D score may seem to experience greater improvements in HRQoL after these procedures. Additionally, it underscores that patient-reported outcome measures are subjective tools and may not entirely mirror a patient's HRQoL. Ultimately, scores can be influenced by factors such as patient mood, recall of events, and rehabilitation experiences [60]. Despite some imperfections, EQ-5D is globally used for measuring HRQoL and has demonstrated strong reliability and validity [61, 62]. Moreover, various other patient characteristics including sex, presence of comorbidities, employment status, living circumstances, and presence of physical

barriers were identified to have significant impacts on EQ-5D change [22]. This further highlights the multifactorial nature of EQ-5D change following hip and knee arthroplasty.

Multivariate analyses of various factors influencing EQ-5D indicated being female and not having comorbidities exhibited positive associations with EQ-5D improvement [22]. One study in female patients undergoing hip or knee arthroplasty experienced higher postoperative HRQoL at 6-months and lower pain levels [63]. This was attributed to a potential increased pain tolerance in women, leading to a perceived higher HRQoL. In contrast, other studies have found a negative association with female sex, residual pain and stiffness after knee arthroplasty [64, 65]. However, McKeon et al. found that both males and females exhibited comparable postoperative functional outcomes [66]. Female sex had a protective effect against complications after hip or knee arthroplasty, but it remained a risk factor for non-home discharge [66]. Regarding comorbidities, a study reported that an increased number of comorbidities was associated with worsened pain and physical function, along with a lower quality of life [67]. Similarly, patients with a higher number of comorbidities were more vulnerable to deterioration in physical function and pain outcomes, coupled with an increased rate of complications and mortality following hip arthroplasty [68]. This suggests that assessing comorbidities in patients undergoing joint replacement can provide valuable insights when speculating postoperative outcomes, including potential changes in the patient's HRQoL.

### Limitations

This study has several limitations. Firstly, there is considerable variability in age group categorization and follow-up time points across the included studies. This introduces potential heterogeneity in the analysis of HRQoL changes amongst older adults; as a result, a meta-analysis of HRQoL changes was not possible. Additionally, variability of sample may extend to factors such as HRQoL, severity of joint damage, and postoperative rehabilitation.

### Conclusion

In conclusion, we found that the young-old and old-old patients exhibited comparable improvements in HRQoL following hip and knee arthroplasty. Older patients did not have greater postoperative complication, longer hospital LOS, and increased mortality. While chronological age merits consideration in planning for hip and knee arthroplasty, greater emphasis should be placed on assessing comorbidities and functional status. This approach addresses the complexities of aging and highlights that older age alone should not be a contraindication for hip and knee arthroplasty. Evaluating biological age, frailty, and overall health provides a more accurate prediction of postoperative outcomes, ensuring that older adults receive appropriate surgical interventions that enhance their quality of life.

### Supporting information

**S1 File. Contains checklist and all the supporting tables.**
(DOCX)

**S2 File. Full literature search results and exclusion reasons.**
(CSV)

**S3 File. All data extracted from primary research sources.**
(XLSX)

## Author Contributions

**Conceptualization:** Yun Jin Chen, Justine Lau, Frances Chung.

**Data curation:** Yun Jin Chen, Justine Lau, Marina Englesakis.

**Formal analysis:** Yun Jin Chen, Justine Lau, Yasmin Alhamdah, Aparna Saripella.

**Investigation:** Justine Lau.

**Methodology:** Yun Jin Chen, Justine Lau.

**Supervision:** Frances Chung.

**Validation:** Yun Jin Chen, Yasmin Alhamdah, Ellene Yan.

**Writing – original draft:** Yun Jin Chen, Justine Lau.

**Writing – review & editing:** Yun Jin Chen, Justine Lau, Yasmin Alhamdah, Ellene Yan, Aparna Saripella, David He, Frances Chung.

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
