## [Decision Letter · Decision Letter 0]

26 Apr 2024

PONE-D-24-06598Changes in health-related quality of life in young-old and old-old patients undergoing elective orthopedic surgery: A systematic reviewPLOS ONE

Dear Dr. Chung,

Thank you for submitting your manuscript to PLOS ONE. After careful consideration, we feel that it has merit but does not fully meet PLOS ONE’s publication criteria as it currently stands. Therefore, we invite you to submit a revised version of the manuscript that addresses the points raised during the review process.

**ACADEMIC EDITOR: Minor Revision**

We look forward to receiving your revised manuscript.

Kind regards,

Raffaele Vitiello

Academic Editor

PLOS ONE

Journal Requirements:

"I have read the journal's policy and the authors of this manuscript have the following competing interests: FC reports research support from the Ontario Ministry of Health Innovation Grant, ResMed Foundation, University Health Network Foundation, Consultant to Takeda, and STOP-Bang Questionnaire proprietary to University Health Network."

3. We note that this manuscript is a systematic review or meta-analysis; our author guidelines therefore require that you use PRISMA guidance to help improve reporting quality of this type of study. Please upload copies of the completed PRISMA checklist as Supporting Information with a file name “PRISMA checklist”.

Reviewers' comments:

Reviewer's Responses to Questions

**Comments to the Author**

1. Is the manuscript technically sound, and do the data support the conclusions?

Reviewer #1: Yes

2. Has the statistical analysis been performed appropriately and rigorously? 

Reviewer #1: Yes

3. Have the authors made all data underlying the findings in their manuscript fully available?

Reviewer #1: Yes

4. Is the manuscript presented in an intelligible fashion and written in standard English?

Reviewer #1: Yes

5. Review Comments to the Author

Reviewer #1: The study is an exhaustive review of literature on HRQoL among patients (65 yrs +) undergoing hip or knee arthroplasty. The work is well presented and supported by data, however, few minor concerns are there, as listed below:

1. Any specific reason why 65+ years of age was the inclusion criteria and why not include people with lesser age?

2. Although the authors have mentioned, through some studies, that gender/sex has been studied as a parameter; the same was not studied here? Why?

3. Other logical criteria like quality of post-operative care, general health status of the patient, etc. were also not included in this study. why?

4. As mentioned by the authors, not all the studies used in this work had the same age grouping. How then the authors use the data from those studies? did you make any further classification of the available data to fit to current study's age groups? If not, why? and what level of error is expected to occur in your results because of that?

5. The study by Miao(2018) had the smallest number of patients and did not report loss of patients during follow-up. Would the results of the current work be impacted significantly, if Miao (2018) study is discarded from the current work?

6. In Table-5, there is significant drop in postoperative EQ-5D score after 24-months period, as compared to 3-months. Why is that so?

7. Conclusions section may be expanded a little.

6. PLOS authors have the option to publish the peer review history of their article (what does this mean?). If published, this will include your full peer review and any attached files.

Reviewer #1: No

---

## [Author Response · Author response to Decision Letter 0]

28 May 2024

We sincerely appreciate your invaluable insights and feedback on our work, thank you for your kind words as well. We have diligently integrated the feedback into our revised manuscript, which is detailed in our responses below.

1. The inclusion criterion of ≥65 years of age was chosen due to the significant impact of age on health-related quality of life following hip or knee arthroplasty. Projections from the World Population Prospects indicate that the proportion of adults aged 65 and older will reach 1.5 billion by 2050, highlighting the importance of understanding age-related outcomes in this demographic. As the population ages, orthopedic surgeons will increasingly encounter elderly patients electing for joint arthroplasty. While hip or knee arthroplasty offers functional improvements, the benefits for older patients must be weighed against potential age-related complications. By focusing on this age group, we aim to optimize surgical decision-making and patient care for a rapidly growing segment of the population. Including younger patients might dilute the specific insights relevant to older patients, who present distinct clinical challenges and considerations. Some of these explanations were already on P4, L57-71.

2. The focus of our systematic review was to assess how older age impacts patients' health-related quality of life after hip or knee arthroplasty. We did not include a thorough comparison of sex differences because the papers that met our inclusion criteria lacked sufficient data on this parameter. Consequently, a comprehensive analysis of sex-based differences was not feasible. However, recognizing the importance of this aspect, we plan to explore sex differences in outcomes in future research.

3. The majority of the papers that met our inclusion criteria did not provide consistent reports on the quality of post-operative care or the general health status of patients. As a result, we could not make comparisons on these factors in our review. Our secondary objectives focused on adverse patient outcomes that were more consistently evaluated in the included papers, such as postoperative complications, length of stay, and mortality.

4.Although consensus is lacking on the optimal sub-classification of older adults into age groups, our study adhered to the convention of designating individuals aged 65-74 years as "young-old" and those aged 75 and above as "old-old." In cases where these age ranges were not employed, we classified individuals under 80 years as "young-old" and those 80 years and older as "old-old." (P9, L178-184) 

This approach introduced variability into our analysis of changes in health-related quality of life among older adults, precluding a meta-analysis. Instead, we conducted a qualitative analysis, recognizing general trends and noting the specific age classifications utilized in each source. 

Although this method introduced variability, we aimed to minimize classification errors by meticulously documenting age groups across studies. While we acknowledge the potential for error resulting from this variability, it was necessary to adapt available data to provide a comprehensive overview. 

5. The exclusion of the study by Miao (2018) would not significantly impact the results of our current work. While the study does support findings reported in other included papers, its unique contribution lies in the incorporation of additional variables such as sex, comorbidities, employment status, living arrangements, and physical barriers at home. The removal of this study would not alter the overall conclusions drawn in our analysis. Nonetheless, its inclusion enriches the discussion by introducing diverse variables for consideration in future research endeavors.

6. The drop in postoperative EQ-5D scores at 24 months, compared to 3 months, likely reflects decreased activity levels in older patients, since the EQ-5D includes measures of mobility and daily activities. While the study did not detail changes in EQ-5D scores over time, the authors reported clinically significant improvement in HRQoL post-operatively for all age groups sustained at all time points. 

The decrease at 24 months may be due to older patients being less active post-arthroplasty, likely related to aging-related physical health and social/employment changes, rather than limitations from the surgery itself. This is supported by the Lower Extremity Activity Scale assessments evaluated in the same study, showing patients aged ≥75 maintained stable activity levels from 12 to 60 months, describing their activity as unrestricted at home and outdoors. In contrast, the <65 years group defined their activity as much higher, including extremely active jobs outside the home, also persistent from 12 to 60 months.

This has now been addressed in our discussion: P23-24, L383-389.

7. This has been expanded: P27, L467-471.

---

## [Decision Letter · Decision Letter 1]

1 Aug 2024

Changes in health-related quality of life in young-old and old-old patients undergoing elective orthopedic surgery: A systematic review

PONE-D-24-06598R1

Dear Dr. Chung,

We’re pleased to inform you that your manuscript has been judged scientifically suitable for publication and will be formally accepted for publication once it meets all outstanding technical requirements.

Kind regards,

Raffaele Vitiello

Academic Editor

PLOS ONE

Additional Editor Comments (optional):

Reviewers' comments:

Reviewer's Responses to Questions

**Comments to the Author**

1. If the authors have adequately addressed your comments raised in a previous round of review and you feel that this manuscript is now acceptable for publication, you may indicate that here to bypass the “Comments to the Author” section, enter your conflict of interest statement in the “Confidential to Editor” section, and submit your "Accept" recommendation.

Reviewer #1: All comments have been addressed

Reviewer #2: All comments have been addressed

2. Is the manuscript technically sound, and do the data support the conclusions?

Reviewer #1: (No Response)

Reviewer #2: Yes

3. Has the statistical analysis been performed appropriately and rigorously? 

Reviewer #1: (No Response)

Reviewer #2: Yes

4. Have the authors made all data underlying the findings in their manuscript fully available?

Reviewer #1: (No Response)

Reviewer #2: Yes

5. Is the manuscript presented in an intelligible fashion and written in standard English?

Reviewer #1: (No Response)

Reviewer #2: Yes

6. Review Comments to the Author

Reviewer #1: (No Response)

Reviewer #2: introduction is clear and comprehensive , search quality is well developed and it summarized other studies in tables .however they should Consider adding a table summarizing the HRQoL outcomes for the different age groups across the studies. This would allow for easier comparison between studies.

7. PLOS authors have the option to publish the peer review history of their article (what does this mean?). If published, this will include your full peer review and any attached files.

Reviewer #1: **Yes: **Dr Gaurav Saini

Reviewer #2: No

---

## [Editor Report · Acceptance letter]

12 Aug 2024

PONE-D-24-06598R1 

PLOS ONE

Dear Dr. Chung, 

I'm pleased to inform you that your manuscript has been deemed suitable for publication in PLOS ONE. Congratulations! Your manuscript is now being handed over to our production team.

Kind regards, 

on behalf of

Dr. Raffaele Vitiello 

Academic Editor

PLOS ONE